# The diversity of floral temperature patterns, and their use by pollinators

Michael JM Harrap[1], Sean A Rands[1]*, Natalie Hempel de Ibarra[2], Heather M Whitney[1]

[1]School of Biological Sciences, University of Bristol, Bristol, United Kingdom; [2]Centre for Research in Animal Behaviour, School of Psychology, University of Exeter, Exeter, United Kingdom

**Abstract** Pollinating insects utilise various sensory cues to identify and learn rewarding flower species. One such cue is floral temperature, created by captured sunlight or plant thermogenesis. Bumblebees, honeybees and stingless bees can distinguish flowers based on differences in overall temperature between flowers. We report here that floral temperature often differs between different parts of the flower creating a temperature structure or pattern. Temperature patterns are common, with 55% of 118 plant species thermographed, showing within-flower temperature differences greater than the 2°C difference that bees are known to be able to detect. Using differential conditioning techniques, we show that bumblebees can distinguish artificial flowers differing in temperature patterns comparable to those seen in real flowers. Thus, bumblebees are able to perceive the shape of these within-flower temperature patterns. Floral temperature patterns may therefore represent a new floral cue that could assist pollinators in the recognition and learning of rewarding flowers.

DOI: https://doi.org/10.7554/eLife.31262.001

*For correspondence:
sean.rands@bristol.ac.uk

Competing interests: The authors declare that no competing interests exist.

## Introduction

Many flowering plants require pollen transport by animals to ensure reproductive success (*Ollerton et al., 2011*). These pollinating animals are often insects (*Kevan and Baker, 1983*), such as bees. To encourage pollinator visits flowering plants create floral displays (*Raguso, 2004*; *Leonard et al., 2012*) which produce diverse floral cues in different sensory modalities (*Kevan and Lane, 1985*; *Bhagavan and Smith, 1997*; *Whitney et al., 2009*; *Hempel de Ibarra and Vorobyev, 2009*; *von Arx et al., 2012*; *Lawson et al., 2017b*). These signals allow pollinators to find and locate flowers (*Spaethe et al., 2001*; *Chittka and Spaethe, 2007*), and also allow pollinators to learn and recognise them (*Heinrich, 1979*; *Raine and Chittka, 2008*). Bees and other pollinators adjust their foraging behaviour to favour visits to more rewarding species found in their environment (*Heinrich, 1979*), avoiding 'mistake visits' to less rewarding flowers in order to enhance their foraging success (*Raine and Chittka, 2008*). Similarly, a floral display that is easily learnt and distinguished from others in its environment ensures greater visitation to the flower (*Galen and Newport, 1988*; *Lynn et al., 2005*) and thus greater reproductive success (*Ashman et al., 2004*; *Bell et al., 2005*; *Schiestl and Johnson, 2013*). Identifiable floral cues are therefore critical to both plant and pollinator.

One flower cue bees can use to recognise flowers is floral temperature (*Whitney et al., 2008*; *Hammer et al., 2009*; *Norgate et al., 2010*). Warming of flowers can occur due to floral thermogenesis (*Seymour and Schultze-Motel, 1997*; *Seymour and Matthews, 2006*; *Seymour et al., 2009*), but is more frequently the result of captured solar radiation (*Totland, 1996*; *Sapir et al., 2006*; *Rejšková et al., 2010*; *Zhang et al., 2010*; *Atamian et al., 2016*). The absorption of sunlight and heat loss is influenced by pigmentation (*Kay et al., 1981*; *Sapir et al., 2006*), structure

**eLife digest** Bees experience the world in a different way to humans. The plants that they visit exploit the bee's senses to make sure that a searching bee can easily find, handle and pollinate flowers. For example, bumblebees can learn to choose between flowers that are different temperatures, using heat as a way of identifying the best flowers.

Some wild flowers are warmer than others when they grow in their natural environment. Recent advances in technology mean that scientists are now able to take a more detailed look at flower temperature than ever before. Harrap et al. used this technology to look at 118 species of plant, including daisies, rockroses and poppies.

Over half of the plants examined had flowers with complex patterns of heat across their petals, echoing the colourful patterns that we see with our own eyes. On average, some parts of the petals were 4–5°C warmer than the rest. In further experiments, artificial flowers that replicated these patterns showed that bumblebees are able to tell apart flowers with different temperature patterns across their petals.

These newly discovered floral heat patterns appear widespread in nature. It is likely that these patterns are a hidden signal to pollinators that, together with other cues like colour and scent, attracts them to the flowers and helps them locate any reward, like nectar.

As well as opening up a new field of research in understanding the interactions between plants and their pollinators, these findings are potentially important given current concerns about climate change. If pollinators are partly reliant on subtle differences in temperature across the surface of a petal, then even small changes in the temperature of the environment could have a large and unanticipated influence on how efficient bees and other pollinators are when they are visiting flowers with hidden heat patterns.

DOI: https://doi.org/10.7554/eLife.31262.002

(*Rejšková et al., 2010*; *Whitney et al., 2011*) and heliotropism (*Totland, 1996*; *Zhang et al., 2010*; *Atamian et al., 2016*), all of which will contribute to how much a certain flower will heat up in given conditions. This can create differences in temperature between different flower species (*Rejšková et al., 2010*; *Kovac and Stabentheiner, 2011*). Using thermal detectors in their antennae and tarsi (*Heran, 1952*), bumblebees (*Dyer et al., 2006*; *Whitney et al., 2008*), honeybees (*Hammer et al., 2009*; *Kovac and Stabentheiner, 2011*) and stingless bees (*Norgate et al., 2010*) can distinguish flowers based on differences in overall temperature. Greater differences in temperature between flowers appear to be easier for bees to detect (*Hammer et al., 2009*), although bees have been shown to be able to detect differences in temperature as little as 2°C (*Heran, 1952*). Floral temperature can also function as a floral reward by keeping pollinators warm while they feed (*Rands and Whitney, 2008*; *Herrera, 1995*). Warmer flowers help insect visitors maintain a body temperature above their minimum threshold for flight (*Heinrich, 1979a*; *Heinrich, 1979c*). This allows pollinators to forage and collect nectar in colder conditions (*Herrera, 1995*), and avoid the metabolic costs they might incur if they have to warm themselves for flight (*Rands and Whitney, 2008*). Therefore, floral temperature cues are likely to be salient to insect pollinators.

As well as being sensitive to differences between the flower and its environment (*Whitney et al., 2008*; *Hammer et al., 2009*), insects should also be sensitive to differences within a floral display. When flowers are observed using infrared thermography (thermal imaging), it is apparent that floral temperature is not necessarily distributed uniformly across the flower surface (*Rejšková et al., 2010*; *Dietrich and Körner, 2014*; *Ladinig et al., 2015*; *Atamian et al., 2016*). It has not been investigated whether any pollinators can learn to recognise flowers based on which parts of the flower are hotter or colder, which will determine the flower's temperature pattern. Understanding whether pollinators can detect temperature patterns within the flower will improve our understanding of how pollinators interact with flowers, and how floral displays have evolved.

In this study, we investigate the capacity of these floral temperature patterns to function as a floral cue. We demonstrate floral temperature patterns are common by taking thermographs of the displays of 118 plant species, that are visited by a range of pollinator groups and show a variety of flower forms, under good weather conditions. We further ask whether bumblebees, frequently a

generalist pollinator group (*Heinrich, 1976*; *Williams, 1989*; *Goulson et al., 2005*), can learn to distinguish rewarding from non-rewarding artificial flowers, based solely on temperature pattern differences comparable to those observed in real flowers.

## Results

### Diversity of floral temperature patterns

The thermographs of flowers of 118 species in different taxa reveal the variety of temperature patterns of different shapes, sizes and locations that pollinators may encounter (*Figure 1* and *Supplementary file 1*). Some species had little to no detectable temperature differences across their surface, for example *Dahlia coccinea* and *Pelargonium echinatum* (*Figure 1*). However, most species observed showed some part of the flower that differed in temperature from the rest of the flower, thus displaying a temperature pattern (*Figure 1* and *Supplementary file 1*). Most often there was a temperature contrast between the flower centre and its periphery, although the extent and shape of contrasting regions varied greatly. In such cases the centre of the flower was often hotter, as in *Bellis perennis* and *Geranium psilostemon*, (but not always, as with *Papaver rohoeas* or *Hydrangea macrophylla*) (*Figure 1*). Warming or cooling of a protruding section of the flower, such as 'landing pad' petals of zygomorphic flowers such as *Crinum*, or the reproductive structures of *Papaver* (*Figure 1*), also frequently created contrasting regions of temperature. Flowers of all sizes showed temperature patterns, such as the large *Hermerocallis* 'autumn red' and small *Bellis perennis* flowers (*Figure 1*).

Of the 118 species thermographed, 65 species (55%) showed within-flower temperature differences of at least 2°C (*Supplementary file 1*). So more than half the species observed show temperature contrasts which at least bees would be able to detect (*Heran, 1952*). Within these 65 species the average temperature difference was 4.89°C ± 2.28 (mean ± SD). While the temperature patterns that can vary greatly between species, we must determine whether pollinators can use such differences in temperature patterns to inform foraging in order to show these differences can be used as floral cues.

### Bumblebees discriminate between flowers with different temperature patterns

We carried out two conditioning experiments investigating the ability of bumblebees to detect temperature patterns. Bumblebees are an appropriate choice of pollinator for investigating whether any pollinators can respond to the observed diversity of temperature patterns. Many bees are generalist pollinators (*Waser et al., 1996*; *Fenster et al., 2004*), and it is known that generalist bees will visit many flower forms and families (*Heinrich, 1976*; *Heinrich, 1979a*; *Williams, 1989*; *Fenster et al., 2004*; *Fontaine et al., 2008*; *de Vere et al., 2017*). Bees also visit flowers which they may not pollinate to carry out larceny (*Inouye, 1980*; *Manning et al., 2002*; *Castellanos et al., 2004*; *Fenster et al., 2004*). There is great variation in size and tongue length both within and between bumblebee species, with long tongued species tending to be specialist, and shorter tongued bumblebees (such as *Bombus terrestris*) tending to be generalist (*Heinrich, 1976*; *Heinrich, 1979a*; *Williams, 1989*; *Goulson et al., 2005*). Bumblebees also occur all over the globe (*Heinrich, 1979a*). Thus, bumblebees, both as individual species and as a large functional group, will experience a large portion of the diversity of floral temperature patterns observed in our survey. This includes some of the species with the most contrasting temperature patterns, and flowers showing near to no temperature pattern at all (*Williams, 1989*; *Goulson et al., 2005*; *Larsson, 2005*; *Fontaine et al., 2008*; *Smith, 2010*). Additionally, the temperature sensitivities of bumblebees are understood better than many other pollinators (*Heinrich, 1979a*; *Dyer et al., 2006*; *Rands and Whitney, 2008*; *Whitney et al., 2008*) and techniques for bumblebee conditioning experiments used here are well established (*Dyer and Chittka, 2004*; *Raine and Chittka, 2008*), making them ideal for investigating pollinator responses to floral displays.

In each of the two experiments bumblebees *B. terrestris* were presented with artificial flowers, either small (40 mm in diameter) or large size (85 mm in diameter) depending on experiment (*Figure 2a and b*). The two experiments with different sized flowers allow us to determine the impact of the size of temperature patterns on the identification of rewarding flowers. By using electrical heating elements, we were able to present differing temperature patterns on both sets of

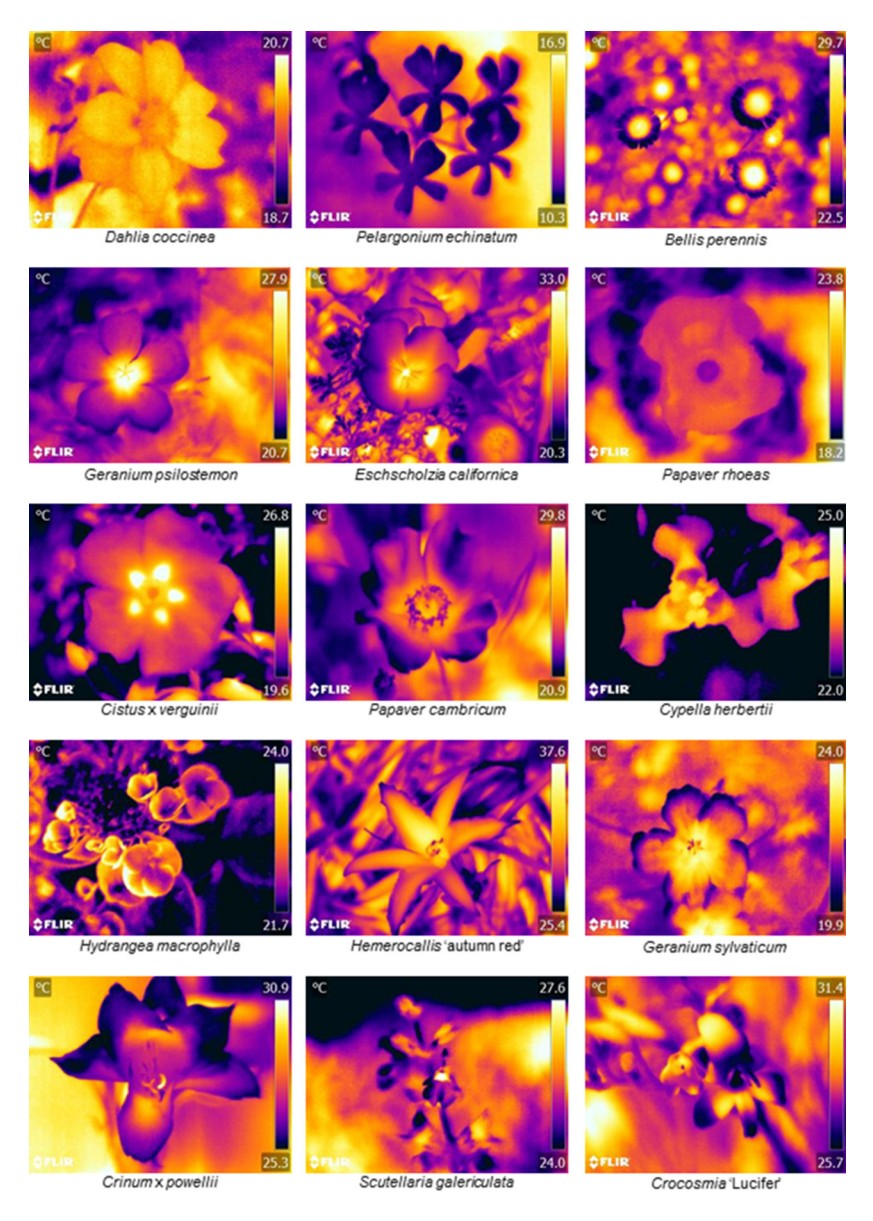

**Figure 1.** Floral thermographs demonstrating the range of floral temperature patterns observed. Floral thermographs demonstrating the range of floral temperature patterns observed. Colour indicates temperature in °C as indicated on the scale bar to the right of each panel. The flower species is labelled below each thermograph. Human colour images corresponding to each thermograph are available in *Figure 1—figure supplement 1*. *Supplementary file 1* summarises the temperature differences measured across all 118 species observed, and thermographs of each species can be found in Supplementary Data available at https://doi.org/10.5061/dryad.qp244 (*Harrap et al., 2017*).

DOI: https://doi.org/10.7554/eLife.31262.003

The following figure supplement is available for figure 1:

**Figure supplement 1.** Human colour images of each flower species shown in *Figure 1*.

DOI: https://doi.org/10.7554/eLife.31262.004

artificial flowers. On each flower size, these temperature patterns had two variants in layout and shape, but did not differ in either overall flower temperature, within-flower temperature contrast, or total area heated, to exclude other means by which bees could recognise variants. Small artificial flowers produced temperature patterns where either the edges of the flower's lid were hotter than the rest of the flower (the 'circle pattern'), or a rectangular section across the middle of the centre of the flower's lid was hotter than the rest of the flower, (the 'bar pattern') (*Figure 2c*). These circle-

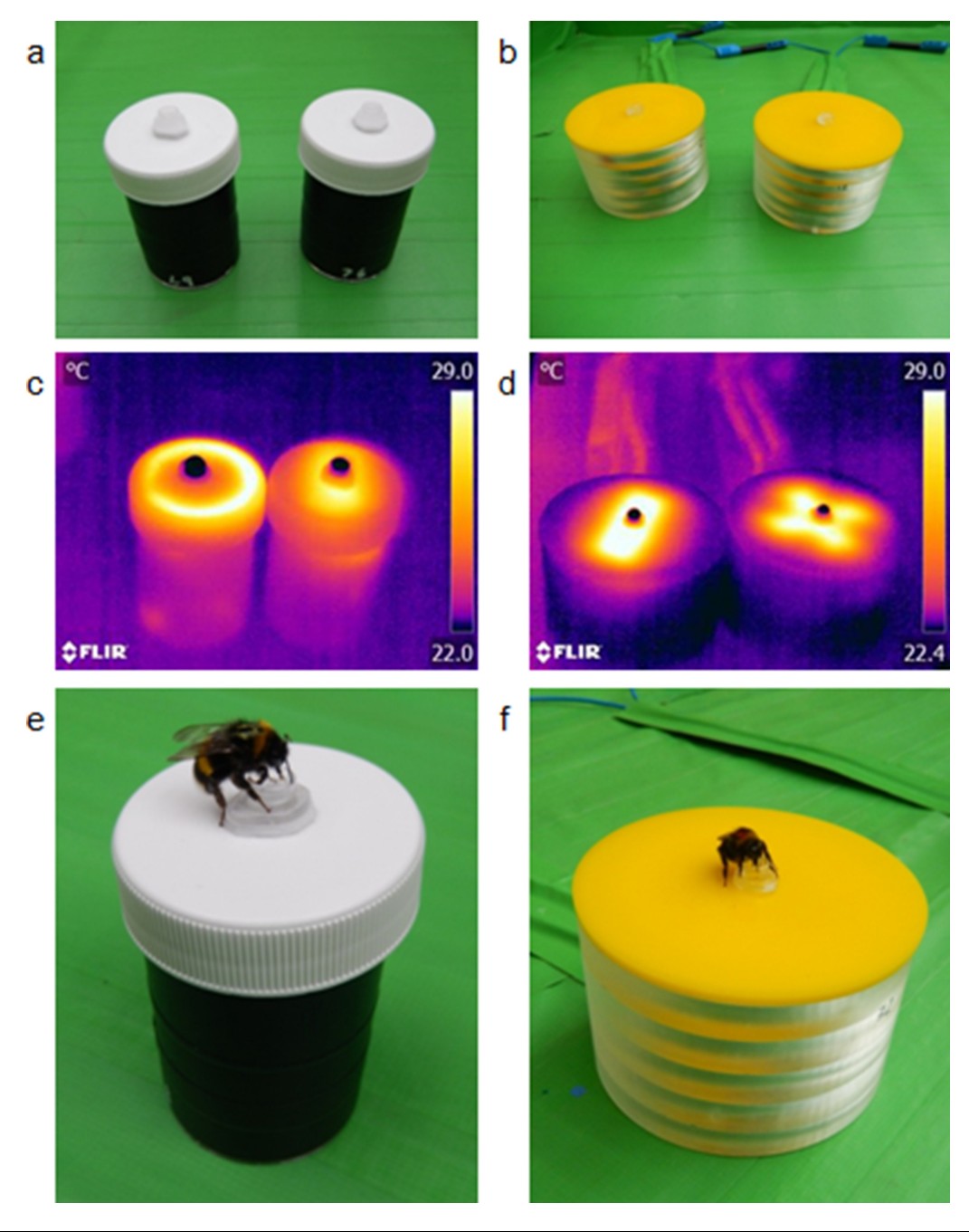

**Figure 2.** The artificial flowers used in the bumblebee learning experiments. Panels a and b: how both variants of artificial flowers used in the small (**a**) and large (**a**) artificial flower experiments appear. Panels c and d: thermographs of both artificial flower variants in in the small (**c**) and large (**d**) flower experiments, demonstrating how artificial flowers within each experiment differ in temperature patterns but not visually. Panels e and f: bumblebees feeding from the small (**e**) and large (**f**) artificial flowers.
DOI: https://doi.org/10.7554/eLife.31262.005

The following figure supplement is available for figure 2:

**Figure supplement 1.** The heating elements stuck to the underside of the small (panel a) and large (panel b) artificial temperature flowers used in the conditioning experiments.
DOI: https://doi.org/10.7554/eLife.31262.006

and bar-shaped temperature patterns were comparable to those displayed by real flowers: flowers with colder centres and hotter peripheries, such as *Papaver rhoeas* and *Hydrangea macrophylla* (*Figure 1*), relating to the circle pattern; flowers with hot centre and colder periphery, such as *Bellis perennis*, *Geranium psilostemon* or *Eschscholzia californica* (*Figure 1*) relating to the bar pattern. Consequently, the differences in temperature patterns between small artificial flowers reflected a large aspect of temperature pattern diversity. Both of the large artificial flower variants used had hotter flower centres: one where the heated parts radiated out from the centre in a 'cross pattern', and one where heated parts spanned across the flower centre in another 'bar pattern' (*Figure 2d*). These larger temperature patterns are similar to those of flowers with hotter centres but differ in the size and shape of the hotter regions of which there are several (compare varieties of *Cistus* and *Geranium*, *Figure 1* and supplementary materials).

Flower naïve bumblebees, *Bombus terrestris audax*, were allowed to visit artificial flowers which provided a drop of sucrose solution (rewarding flowers), or water (nonrewarding flowers), in the centre of the flower hidden in a small well (*Figure 2e and f*). There were three test groups: (i) 'Bar rewards' group where the bar temperature pattern was rewarding, and the distractor pattern nonrewarding (cross in large or circle in small flowers); (ii) 'Circle/cross rewards' group where reciprocally the circle or cross temperature pattern was rewarding, and the distractor nonrewarding (bar pattern); (iii) 'Control' group where heating elements in the flowers were disconnected, and thus neither rewarding or nonrewarding flowers showed temperature patterns. The relationship between foraging success (probing or feeding from flowers rewarding with sucrose solution, as well as not probing when visiting on nonrewarding flowers offering water) and the experience bees had of the flowers (number of flower visits the bees made) was compared between the three test groups.

When foraging on small artificial flowers, bumblebees learnt to identify rewarding flowers when they differed in temperature patterns (*Figure 3*), but did not learn in the control group. When models of bumblebee foraging success in the learning phase were compared, the relationship between

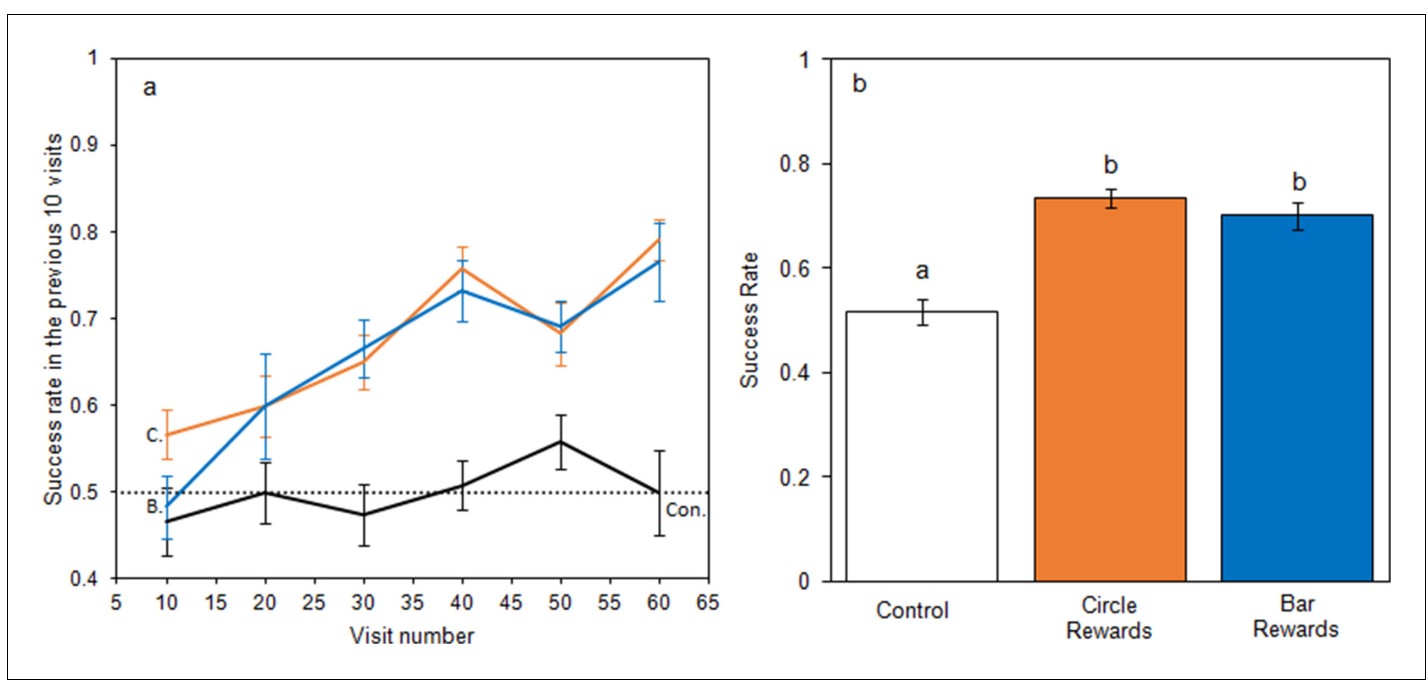

**Figure 3.** Bumblebee learning within our small artificial flower experiment. Panel a: the relationship between bees' foraging success and experience of the small artificial flowers (flower visits made) during the learning phase. The dotted line indicates the 50% success level. Solid lines indicate the mean foraging success of bees in the previous 10 visits. Error bars represent ± SEM. Colour and label of solid lines and error bars correspond with test group: black, the control group, labelled 'Con.'; orange, Circle rewards group, labelled 'C.'; blue, Bar reward group labelled 'B.'. Panel b shows mean foraging success ± SEM of bees in different test groups during the nonrewarding test phase. Letters above bars denote groups as defined by *post hoc* Tukey's tests where p < 0.05. 12 bees completed this experiment in each test group (36 bees in total from four different nests).
DOI: https://doi.org/10.7554/eLife.31262.007

success and experience varied between test groups (*Figure 3a*), with models that allowed test group to have an interacting effect with experience producing a lower AIC (−226.3 *vs.* −216.9, ΔAIC = 9.4) and a better fit (Δdeviance = 13.4, df = 2, p < 0.01) than models that did not. Bees from the control group foraged randomly throughout the experiment maintaining a 50% success rate, experience having no effect on success (AIC −83.3 *vs.* −83.0 ΔAIC = 0.3; Δdeviance = 1.6, df = 1, p = 0.201). When flowers differed in temperature pattern, bees began with a success rate comparable to the control group but improved with experience; this occurred regardless of which temperature pattern corresponded with rewards (Circle rewards: AIC −92.1 *vs.* −68.5, ΔAIC = 23.6; Δdeviance = 25.6, df = 1, p < 0.001. Bar rewards: AIC −50.5 *vs.* −28.0 ΔAIC = 22.5; Δdeviance = 24.5, df = 1, p < 0.001). When the conditioned preference was tested in nonrewarding tests, bees in the bar and circle reward groups made more correct visits than the control group ($F_{2,33}$ = 23.8, p < 0.001, *Figure 3b*). These results demonstrate that bumblebees can learn and alter foraging decisions based on differences in temperature patterns.

Bumblebees also showed the ability to perceive temperature patterns in large-sized flowers (*Figure 4*), although test groups showed similar shaped relationships between success and experience. Models including an interaction between test groups and those that did not, were comparable in terms of AIC (*Richards, 2008*) (AIC −290.88 *vs.* −287.72 ΔAIC = 3.16), but were a better fit (Δdeviance = 7.16, df = 2, p = 0.03). Nevertheless, which test group bees were in still had a significant effect on the level of success achieved (AIC −287.72 *vs.* −266.71 ΔAIC = 21.01, Δdeviance = 25.01, df = 2, p < 0.001), with Bar and Cross reward groups achieving a greater level of success than the control. Thus, the presence of temperature patterns improved bumblebee foraging success, indicating their ability to use these larger patterns to distinguish flowers.

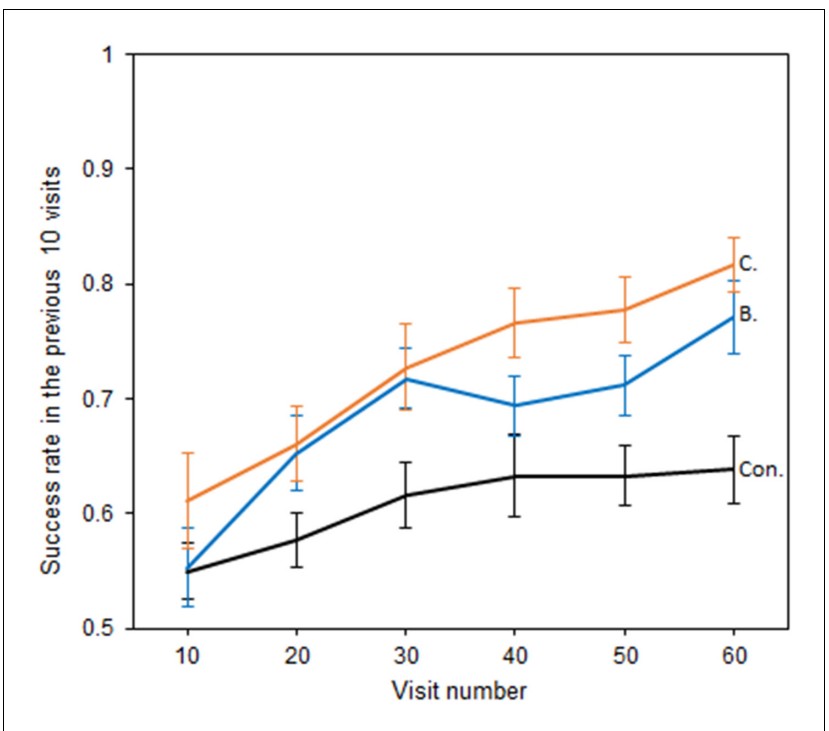

**Figure 4.** The relationship between bumblebees' foraging success and experience of the large artificial flowers (flower visits). Solid lines indicate the mean foraging success of bees in the previous 10 visits. Error bars represent ±SEM. Colour and label of solid lines and bars correspond with test group: black, the Control group, labelled 'Con.'; orange, Cross rewards group (rewarding cross pattern), labelled 'C.'; blue, Bar reward group (rewarding bar pattern), labelled 'B.'. 18 bees completed this experiment in the control and Cross rewards group and 17 bees in the Bar rewards group (53 bees in total from seven different nests).
DOI: https://doi.org/10.7554/eLife.31262.008

The increase in success rate in the control group in this experiment (unlike in the previous small flower experiment) can be explained by the experimental set-up leading to spatial preferences within the arena that developed during training. Three rewarding and three nonrewarding flowers were present in the arena during the large flower experiment due to space constraints, and there was a reduced ability for random re-arrangement of each flower due to wiring constraints. Bees have a great capacity for spatial learning (*Burns and Thomson, 2006*; *Robert et al., 2017*), and therefore the control group may have learnt to identify within each foraging bout which regions of the arena contained more rewarding flowers. However, even with this spatial learning, the presence of temperature patterns improved bumblebee foraging success on the large artificial flowers.

## Discussion

The results of both of the conditioning experiments showed that temperature pattern differences improved the ability of bumblebees to distinguish between rewarding and nonrewarding artificial flowers (*Figures 3* and *4*). This suggests that floral temperature patterns can function as a floral cue. The main variation observed in floral temperature patterns were between flowers with hot centres and cold edges and *vice versa* (see *Figure 1* and *Supplementary file 1*), and bees foraging on the small artificial flowers were observed to be able to distinguish similar differences (*Figure 3*). Furthermore, bees foraging on large artificial flowers could distinguish between two differently-shaped patterns where the centre of the flower was hotter (*Figure 4*), demonstrating that bumblebees can detect more detailed aspects of temperature signals. Artificial flowers showed within-flower temperature differences similar to that of real flowers (*Figure 2* and *Supplementary file 1*). Real flowers can show a greater degree of variety in the temperature differences than those used in our experiments, which represented flowers showing the clearest temperature patterns (*Supplementary file 1*). However, bees have been shown to have a high sensitivity to differences in temperature (*Heran, 1952*; *Dyer et al., 2006*) and are therefore likely to detect the lower temperature differences as well as the higher. The use of floral temperature cues might not be limited to bumblebees, since other pollinating insects have been observed to detect and respond temperature differences between different flowers (*Sapir et al., 2006*; *Kleineidam et al., 2007*; *Hammer et al., 2009*; *Kovac and Stabentheiner, 2011*), and therefore may also be able to use temperature patterns as cues. Furthermore, it did not appear that temperature patterns were limited to flowers associated with bumblebees. Temperature patterns appear to be a floral phenomenon, rather than a 'bee flower' phenomenon. Several of the Asteraceae (which are known to be visited by a variety of insects including bumblebees, *Goulson et al., 2005*), as well as primarily bee pollinated flowers such as *Eschscholzia californica* (*Smith, 2010*), were among those that produced the most contrasting temperature patterns (*Supplementary file 1*). However, other plants attracting similar pollinators were also observed to produce little temperature contrast across their surface. Additionally, some plants associated with moths and hummingbirds, such as *Crinium* and *Crocosmia* (*Manning et al., 2002*; *Goldblatt and Manning, 2006*), were also observed to produce contrasting temperature patterns (*Supplementary file 1*).

Demonstrating that floral temperature patterns could present a floral cue raises the question as to how they might be generated, and there are several potential mechanisms. Different flower species differed in which structures generated temperatures patterns (*Figure 1* and *Supplementary file 1*). Some patterns are created by hotter or colder parts of the petals, and others by hotter or colder reproductive structures. The variation in shape and contrast of temperature patterns between different plants derived from the same species (*i.e.* cultivars, subspecies) suggest that small changes in floral characters can influence temperature patterns. This is perhaps most evident in the various *Cistus*, *Gazania* and *Knautia* flowers thermographed (*Supplementary file 1*). Floral morphology appears to influence temperature pattern generation, as structures in a position more likely to capture light tended to be warmer (e.g. the exposed petals in the landing pad of *Crinum*). Structures that were more densely packed, and might retain heat better were often warmer (such as the florets of composite inflorescences). Likewise, colour differences in the visible spectrum often appeared to occur alongside temperature differences (*Figure 1* and *Figure 1—figure supplement 1*). Such observations are in agreement with our understanding of the influence of solar radiation (*Totland, 1996*; *Sapir et al., 2006*; *Rejšková et al., 2010*; *Kovac and Stabentheiner, 2011*) and floral structure (*Miller, 1986*) on floral temperature. Additional potential influences on temperature include floral

metabolism (*Seymour and Schultze-Motel, 1997*; *Seymour and Matthews, 2006*), active heat loss by transpiration (*Gates, 1968*; *Tsukaguchi et al., 2003*) and petal epidermal cell shape effects (*Whitney et al., 2011*). Further study of how these influences differ across the floral surface will help us gain a greater understanding of floral temperature pattern generation.

Ecological factors might influence a flower's capability to generate temperature patterns pollinators can detect. The amount of sunlight captured limits floral warming in non-thermogenic plants (*Totland, 1996*; *Rejšková et al., 2010*; *Zhang et al., 2010*). While *Rejšková et al., 2010* found that artificially shaded *Bellis perennis* flowers maintained temperature patterns, overall temperature of the flower and temperature contrast between regions decreased, and shaded *Anemone nemorosa* cooled to even temperature across the flower. Pollinators may only be able to use temperature pattern cues during sunny weather and when flowers grow in open non-shady environments. Understanding how floral temperature patterns change with environmental conditions, and the sensitivity of pollinators to changing temperature patterns (including how small a contrast in temperature that pollinators are able to detect), will reveal the level of influence that environmental factors have on temperature patterns. It may be that flowers that grow in less sunny climates and in shadier habitats may not be under strong selection to produce complex thermal cues such as temperature patterns. Plants in these conditions may seldomly generate temperature patterns (*Rejšková et al., 2010*), and pollinators may not be able to detect or respond to these patterns. Several of the flower species that produced the greatest contrasts in temperature within the flower are associated with hot and dry climates (*e.g. Osteospermum* and *Dimorphotheca* species) or with more open environments (*e.g. Geranium psilostemon* and *Eschscholzia californica*, *Supplementary file 1*), even though all samplings took place in similar conditions. This may reflect such plants experiencing greater selection to produce thermal cues.

Flowers are multimodal displays - they produce many different kinds of cues simultaneously (*Raguso, 2004*; *Leonard et al., 2012*), despite pollinators often being able to distinguish flowers based on a single cue (*Bhagavan and Smith, 1997*; *Dyer and Chittka, 2004*; *Clarke et al., 2013*). The benefits of this multimodality are only just starting to be understood (*Leonard et al., 2012*; *Kaczorowski et al., 2012*; *Leonard and Masek, 2014*; *Lawson et al., 2017b*; *Lawson et al., 2017a*). 'Novel' sensory cues, such as floral electrostatic fields, have been found to be equally beneficial in foraging maintaining accuracy (*Clarke et al., 2013*). The discovery of another floral cue that bumblebees can use to recognise flowers, temperature patterns, encourages further investigation into this apparent redundancy in floral signalling and the potential benefits multimodal signalling confers. The frequent overlap of temperature patterns with structural and visual elements of the floral display perhaps makes them ideal for investigation of how floral signals interact within multimodal displays.

Thermal imaging of floral temperature reveals that flowers show a diversity of temperature patterns. It is known that bees can distinguish differences in temperature between flowers (*Whitney et al., 2008*) and using temperature as a reward (*Rands and Whitney, 2008*), and we have shown here that bumblebees can use these floral temperature patterns as a cue to recognise flowers and make informed foraging choices based upon them. This ability does not seem to be influenced by the size of the flower and its floral temperature pattern. Thus, floral temperature patterns may be added to the growing number of floral cues (*Raguso, 2004*; *Leonard et al., 2012*) that pollinators, at least bumblebees, may be able to utilise to identify more rewarding flowers in their environment.

## Materials and methods

### Sampling of floral temperature patterns

Thermographs of floral blooms (flowers or flowering heads) were taken in Royal Fort Gardens and the University Botanic Garden, Bristol and in the National Botanic Garden of Wales, Carmarthen. Species were selected with the aim of sampling flowers visited by a wide range of floral visitor groups and as broad range of floral shapes, colours and phylogeny as possible. Due to thermal camera limitations in minimum area of measurement (*I.T.C, 2008*; *Usamentiaga et al., 2014*) very small flowers, when not part of a compound inflorescence, could not be sampled. Cultivars and subspecies were also thermographed. Any additional cultivars and subspecies were counted as the same species as the one they were derived from when calculating temperature pattern occurrence or average

within flower temperature difference. In such cases whichever variant showed the lowest temperature difference was used, providing more conservative estimates. Thermographs were taken on clear and sunny days, or inside a controlled glasshouse with near-UV permeable windows, while in sunlight. Mean ambient temperature for during sampling was 14.3°C (SD 4.7). More details on the weather conditions are available in *Supplementary files 2* and *3*. All thermographs were taken with a FLIR *E60bx* thermal camera (FLIR systems, Inc., Wilsonville, USA), to a standard acceptable for I.T.C. guidelines (*I.T.C, 2008*; *Usamentiaga et al., 2014*). The thermal infrared emissivity was set at 0.98. This value is the estimate for vegetation (*Rubio, 1997*; *López et al., 2012*) and has been used for floral tissue (*Rejšková et al., 2010*; *Dietrich and Körner, 2014*). For the sake of efficiency, reflected temperature was kept at 23°C for all thermographs, due to the high emissivity of floral tissue this would have a minimal effect on temperature readings. All thermographs were viewed and measurements taken using in FLIR tools software (*Flir Systems INC, 2015*). Using the point measurement functions, the temperature differences between the hottest and coldest points on the flower were measured and used to calculate the temperature range across each flower.

## Bumblebee experiments

Established bumblebee differential conditioning techniques (*Dyer and Chittka, 2004*; *Raine and Chittka, 2008*; *Whitney et al., 2008*; *Whitney et al., 2009*; *Clarke et al., 2013*) were used to investigate whether bumblebees could learn to tell apart flowers based on differences in temperature patterns. All experiments were carried out in lab conditions, using flight arenas as described in *Clarke et al. (2013)*. Ambient temperature was maintained at 21°C and flight arenas were ventilated regularly when access hatches were opened to insert artificial flowers. Flower naïve bumblebees, *Bombus terrestris audax*, were supplied by Biobest (Westerlo, Belgium) *via* Agralan (Swindon, UK) or Syngenta-Bioline (Clacton-on-Sea, UK).

## Artificial flowers
### Overview of flower design

Small and large artificial flowers were made from plastic cylinders with an insulated feeding well in the centre of a plastic lid (*Figure 2*). Electric heating elements were stuck to the underside of the lid. In the small flowers, this heating element was made from resistance wire and a pressure sensitive putty (Blu Tack: Bostik, Paris, France) heat sink. In the large flowers, four 1 Ω resistors with a built-in sink were used. In the small flowers, these heating elements could be altered in shape to create two different temperature patterns: a 'circle pattern' about the edge of the lid, and a 'bar pattern' across its centre (*Figure 2c*). Altering the arrangement of the resistors in large flowers created two patterns: a 'cross pattern', where resistors radiated from the flower's centre; and another 'bar pattern', where resistors were equally spaced across the flower's length (*Figure 2d*).

The small artificial flowers were powered by 1.2 V AA batteries wired inside the flower. All small artificial flowers normally reached a temperature of 33°C at the heated parts above the heat sink and 25°C on the parts that were not heated, with temperature differences approximately 8°C. These varied slightly between flowers and with time as flowers heated over the experiment but not consistently between temperature patterns in a way that could inform foraging decisions. Large artificial flowers were wired in series to a variable power unit (voltage ranging from 1.5 V and 15 V). This created a consistent voltage drop across each flower, thus the heating and area heated was the same between patterns. The temperatures of large flowers were monitored during tests using the thermal camera and a pair of flowers outside the arena wired into the same series as those presented to the bees. By varying the voltage temperatures were maintained, at approximately 24°C in cold parts and 30°C in hot parts. The temperature difference was maintained between 5°C and 7°C. Static electric signals generated by the larger artificial flowers were checked using a non-contact voltmeter and found to be below the 10 V charge that bumblebees can detect (*Clarke et al., 2013*) and thus could not conflate results. As flowers within each experiment had the same heading elements, differing only in the shape, the area heated and the overall temperature of artificial flowers did not differ in a way which could inform bee foraging decisions, only the temperature pattern (*Figure 2*). The temperatures and within flower temperature differences reached by our artificial flowers are above the average values observed in our survey, yet are within the range observed (see *Figure 1* and

Supplementary file 1). The aim of this study is to investigate bee's capacity to detect temperature patterns, thus they represent flowers that show well contrasting temperature patterns.

## Detail of flower construction

Small artificial flowers were built from a specimen jar (Thermo scientific sterilin (Newport, UK), PS 60 ml, with white plastic lids). An upturned 0.5 ml Eppendorf (Hamburg, Germany) tube lid, insulated by a section of 1 mm thick plastic foam, was stuck to the centre of the jar's lid (see *Figure 2a*). This Eppendorf tube lid functions as a feeding well to contain sucrose solution or water but, with the plastic foam, also insulates it from heating. A 13 cm length of 0.32 mm, 17.87 $\Omega$ m$^{-1}$ kanthal resistance wire was cut and 11 cm of this was covered and stuck down to the underside of the feeders lid with blu tack. This left two 1 cm 'leads' on each end of this heating element. Two patterns were created by the blu tack (Bostik, Paris, France). The first, a circle about the rim of the jar's lid, placed in such a way that it was still inside the treading of the jars screw. The lipped design of the jar allowed this to be done easily. In the second, where the wire was folded into an M shape along the centre of the jars lid, the blu tack creating a bar shape. Care was taken for the blu tack not to cover more than 3 cm$^2$ in each temperature pattern. The wires leads were then linked to a single AA battery in a cradle using two cut free sections of a connector block (*Figure 2—figure supplement 1*). When a 1.2 V AA battery was inserted into the cradle, the current begins to heat up the resistance wire thus causing the blue tack to function as a heat sink heating up the top of the flower lid creating a circle or bar shaped temperature pattern depending on the shape of the blu tack heat sink (see *Figure 2c*). As the length of the resistance wire and the battery type was the same in each flower, the amount of heating varied little between flowers (*Figure 2c*). As the area covered by the blu tack was also kept the same between patterns, the area heated was also the same between temperature patterns. This battery in the cradle was placed inside the jar and the lid closed over it. Black electrical tape was wrapped about the outside of the jar to conceal the content from bees and prevent the possibility that bees may visually identify the shape of the blu tack heat sink.

Large artificial flowers were made using an 8 cm yellow cast acrylic disc that was built to slot into an 5 cm tall cylindrical stand. Again, an insulated feeding cup was stuck to the centre top of the disk (see *Figure 2b*). Four 1 $\Omega$ resistors with a built-in heat sink (Welwyn (TT Electronics, Woking, UK), through hole wirewound resistors) were stuck to the underside of the disks with resin. These were arranged in either a cross pattern radiating from the centre of the flower or a bar pattern spaced equally across the underside (*Figure 2—figure supplement 1* panel b). These resistors were wired in series to two long blue insulated copper wires with connectors. These wire leads were covered by a sleeve made of card and green tape to match the floor of the arena and minimise the distraction to the bees. Eight of these artificial flowers were attached to each other again in series, to a variable power supply (ranging between 1.5 V and 15 V). During the experiment, this allowed six artificial flowers to be present in the area and the temperatures of a further two to be monitored outside the area with a IR camera. When the power source was turned on the artificial flower's top heated up above the resistors. This created two patterns of temperature on the flower's top, both hotter in the centre of the flower but differing in shape according to the placement of resistors (*Figure 2d*). As each flower had four resistors in series, all flowers heated up at the same rate and the area heated was the same across all the flowers. Varying the voltage allowed us to control the heating within the flowers. The cylindrical stand of the artificial flowers was transparent but clouded and thick, thus bees were unlikely to be able to see though to recognise flowers by the pattern of resistors.

As the Perspex lid of the flight arena was non-transparent to the thermal infrared radiation that the IR camera detects, a method was needed that allowed researchers, but not bees, to identify the artificial flowers while bees foraged. To allow identification of the temperature pattern in a way humans but not bees would manage, randomly generated even and odd numbers were written on the side of both kinds of artificial flower corresponding with the flowers temperature pattern. These numbers included several digits to allow even and odd digits to occur on all flowers thus bees could not use the presence of the number's shapes to recognise a flower. As jars and cylinder stands could be switched we also were able to change whether even or odd numbers corresponded with rewards (see *Figure 2a and b*).

## Learning experiments

Before bees began foraging they were assigned to one of three test groups described above (Circle/Cross rewards, Bar rewards, Control). This was done with the goal of balancing occurrence of bees from the same nest across test groups, although this was subject to bee activity. An individual bee only foraged in one test group and were not used in both experiments. Both conditioning experiments began with a learning phase, where bees were presented with a choice of flowers placed randomly about the flight arena floor. Bees were allowed to freely forage on the artificial flowers, and return to their nest. This time between a bee departing the nest to forage and returning was classified as a single foraging bout. During the learning phase feeding wells of the rewarding artificial flowers (as determined by the bee's test group) were filled with 25 $\mu$l of 30% sucrose solution and the feeding well of nonrewarding artificial flowers with 25 $\mu$l of water. In small flower tests, sixteen flowers (eight of each temperature pattern) were presented to the bee. In large flower tests, six flowers (three of each temperature pattern) were presented. Typically, bees made contact with the flower top while hovering above it before quitting flight and landing. If a bee landed on the flower it would normally approach the feeding well and extend its proboscis and attempt to feed from the sucrose solution presented in rewarding flowers (*Figure 2e and f*). It could also decide to depart without attempting to feed. As bees detect temperature *via* touch (*Heran, 1952*), physical contact with the top of the flower was considered a landing, even if the bee did not quit flying. Bees were each observed for 60 flower landings. Bees completed the learning phase in 5.69 ± 1.79 and 8.60 ± 2.63 foraging bouts (mean ± SD) for the small and large flower experiments, respectively, making 10.53 ± 6.58 and 6.97 ± 3.96 landings per bout. At each landing, we monitored whether the bee fed from the feeder or left without feeding. For small flower experiments the learning phase was followed by a test phase. In the test phase, bees were allowed to forage freely as discussed above. Here bees were presented with a fresh set of sixteen small temperature pattern flowers with 25 $\mu$l of water in feeding wells but presenting the same temperature patterns, or lack of patterns in control group, the bee had experienced in the training phase. Bees were observed for twenty flower landings in this test phase. A test phase was not carried out in the large flower experiment as the large flowers limited the number that could be sensibly placed within the arena.

In small temperature pattern experiments, flowers were not interfered with by the experimenters while the bee was in the flight arena foraging. This was to minimise disturbance of the foraging bees. Once a bee had emptied the feeder of a flower any subsequent returns to that flower during the same bout were not counted. This was done so that a bee's foraging success was not influenced by encounters with empty feeding wells. It is not possible to distinguish whether a bee withholds its probing response because it is correctly responding to a nonrewarding flower (or incorrectly responding to a rewarding flower) or because the feeding well is empty. In large temperature pattern experiments, flowers were topped up after the bee departed and moved to a different point in the arena, as the small number of flowers meant bees often had to visit flowers more than once in a bout. Return visits were not counted unless the flower had been moved to a different location and refilled whilst the bee was flying elsewhere in the arena. In both experiments after a bee returned to the nest, the end of a foraging bout, all the artificial flowers were taken out of the arena. Flower feeders were emptied and refilled to prevent differences in reward temperature developing. The flower tops were then wiped down with ethanol removing any scent marks the bees could have left. Thus, flowers were cleaned regularly preventing the bee from using these to recognise rewarding flowers. Temperature patterns were then checked with the thermal camera before placing flower feeders back in the arena, replacing any flower that ceased to present the temperature pattern due to a fault.

Each flower landing was classed as correct or incorrect, as described in previous bee conditioning studies (*Whitney et al., 2008*; *Whitney et al., 2009*; *Clarke et al., 2013*). In the learning phase experiments extending their proboscis into the feeding well (probing and/or feeding) on a rewarding flower, or not doing so when landing on a nonrewarding flower, was deemed a correct action. Doing otherwise was deemed incorrect. In the test phase all flowers were non-rewarding, therefore scoring flowers as 'rewarding' and 'nonrewarding' was determined by the reward scheme in the preceding learning phase. So, probing the feeding well of flowers with the temperature pattern that had been rewarding in that bee's test phase, or not probing after landing on a flower showing the temperature pattern that had been non-rewarding were correct actions in the test phase. Success over the

previous 10 visits (starting at visit 10, then 20, 30, *etc.*) in the learning phase and overall success rate in the test phase were calculated for each bee. Comparing foraging success between the control bees and bees that had foraged on flowers with temperature patterns differences allows us to evaluate if temperature patterns aided bumblebee learning.

## Statistical analysis

Data were analysed using *R* 3.1.1 (**R Core Team, 2008**). The success rate data from the learning and test phase underwent an arcsine transformation in order to account for it being bound between 0 and 1. The arcsine of success probability across the whole test phase was compared between the three test groups using analysis of variance. Bee identity was included as a random factor.

Generalised linear model techniques and AIC model simplification were used in our analysis of bumblebee foraging success during the learning phase of our experiments. While differential conditioning data are often analysed by *t*-tests on the first and last 10 visits bees make during learning (**Clarke et al., 2013**), the model simplification technique used here has the advantage of including all visits made throughout the learning phase in comparisons and allows more specific comparisons of the influences on learning between each test group. For this reason, we feel the following model simplification technique is more appropriate and informative for the learning data collected in this study.

Not counting revisits to emptied flowers while scoring foraging success does mean the balance between rewarding and non-rewarding flowers could change as flowers are emptied, especially during the learning phase, as bees are more likely to empty the wells of rewarding flowers. This effect was minimal in the large flower experiments, as flowers were refilled shortly after bees departed from them. In the small flower experiments, there was a much larger number of flowers in the flight arena and bees seldom visited all of them in a bout. Small flowers were refilled at the end of each bout, on average every 10.53 visits. So, any changes in the balance of rewarding and unrewarding flowers did not persist for long. Furthermore, bees can carry out correct foraging actions on rewarding and unrewarding flowers showing probing, or not, as described above. Thus, the capacity of bees to forage correctly does not change as flowers are emptied, as long as some flowers still have sucrose or water in their feeding wells. Consequently, the impact of a changing balance of rewarding and non-rewarding flowers on scoring of pollinator foraging success is likely small and short-lived, thus was not included within our analysis.

The following represents our full model before any simplification was applied:

$$y_{nx} = i + (\ln x * l) + T(s_t + (\ln x * c_t)) + C(s_c + (\ln x * c_c)) + (b_n + (\ln x * r_n)) \tag{1}$$

Where $y_{nx}$ is the arcsine success rate of bee $n$ over the previous 10 visits to the artificial flowers, at $x$ flower visits. $x$ is the number of the visits the bee has made to the artificial flowers, the data for y is calculated in blocks of 10 (10, 20, 30, 40, 50, 60). $i$ is the initial arcsine success rate, the intercept, for bees in the bar rewards test group when $x = 0$. $l$ dictates the change in arcsine success rate with increased $x$ in the bar test group, thus $l$ is effectively the learning speed parameter and allows bee's experience to affect success rate. $T$ and $C$ are Boolean parameters which allow the model to alter $y$ depending on which test group the bee is in. $C$ indicates whether the bee is in the control group, where:

$$C = \begin{cases} 0, & \textit{bee is not in the control group}; \\ 1, & \textit{bee is in the control group}; \end{cases} \tag{2}$$

T indicates whether the bee is in the circle rewards or cross rewards test group, depending on the experiment (see above and main text), where:

$$T = \begin{cases} 0, & \textit{bee is not in the cross or circle test group}; \\ 1, & \textit{bee is in the cross or circle test group}; \end{cases} \tag{3}$$

$s_c$ and $s_t$ are the change in initial arcsine success rate, relative to $i$, for bees in the control and circle or cross test groups respectively. $c_c$ and $c_t$ are the change in learning speed, relative to $l$, for bees in the control and circle or cross test groups respectively. Variation between individual bees was included in our model as a random factor. $b_n$ and $r_n$ represent the change in initial arcsine

success rate and learning speed, for bee number $n$. In the model described in *Equation (1)* parameters $i$, $l$, $s_c$, $s_t$, $c_c$, $c_t$, $b_n$ and $r_n$ are parameters to be estimated.

Model simplification procedure involved paired comparisons between the standing 'best model', beginning with the full model described in *Equation (1)*, with a simpler model. Simpler models were constructed from the standing best model but with further parameters removed (effectively forcing the relevant parameters to equal 0), in the order described below. Should the removal of the parameter has no significant effect on AIC, as laid out by *Richards, 2008*, this simpler model would become the best model for the next comparison. If removal of a parameter led to a significant increase in AIC the standing best (more complex) model would remain the best for the next comparison.

Initially, the effects of random factors were compared, a model without $r_n$ was compared to the complete model. This allowed testing of whether individual bees differed only in intercepts or intercepts and learning speed (as in the full model). In both experiments $r_n$ had no significant effect on the model, and is thus not included in subsequent models below. Secondly interaction effects were investigated by removing $c_c$ and $c_t$. This created a model where the shape of the relationship between $x$ and $y$ in all test groups was dictated only by $l$.

Should the best model according to AIC, find no significant interaction the effects of the test groups would be investigated by removing $T$ and $C$ creating a model where all test groups both showed the same intercepts and learning. Finally, the impact of experience on success was compared by removing the learning parameter $l$. Should the best fitting interaction model include interaction effects individual models for each test group would be fitted as follows:

$$y_{nx} = i + (\ln x * l) + b_n. \tag{4}$$

For each test group, using the model described in *Equation (4)*, we tested whether bee foraging success changed with the number of visits the bee has made by removing $l$.

## Acknowledgements

MJMH was supported by a Natural Environment Research Council studentship within the GW4 +Doctoral Training Partnership [NE/L002434/]. HMW was supported by an ERC Starting Grant (#260920). The authors would like to thank Natasha de Vere, Laura Jones and the National Botanic Garden of Wales for use of their facilities; Nick Wray and the Bristol Botanic Gardens for use of their facilities and assistance with plant species identification; Paul Chappell and Derek Carr for manufacturing the large artificial flowers; and Andy Whitcher and the Infrared Training Centre for training and advice concerning infrared cameras.

## Additional information

### Funding

| Funder | Grant reference number | Author |
|---|---|---|
| H2020 European Research Council | #260920 | Heather M Whitney |
| Natural Environment Research Council | NE/L002434 | Michael JM Harrap |

The funders had no role in study design, data collection and interpretation, or the decision to submit the work for publication.

### Author contributions

Michael JM Harrap, Conceptualization, Data curation, Formal analysis, Investigation, Visualization, Methodology, Writing—original draft, Writing—review and editing; Sean A Rands, Natalie Hempel de Ibarra, Conceptualization, Supervision, Funding acquisition, Methodology, Writing—review and editing; Heather M Whitney, Conceptualization, Resources, Supervision, Funding acquisition, Methodology, Project administration, Writing—review and editing

Author ORCIDs
Michael JM Harrap ⓘ http://orcid.org/0000-0003-0515-2348
Sean A Rands ⓘ http://orcid.org/0000-0002-7400-005X
Natalie Hempel de Ibarra ⓘ https://orcid.org/0000-0002-0859-8217
Heather M Whitney ⓘ https://orcid.org/0000-0001-6450-8266

## Ethics

Animal experimentation: No ethical permission was required for the experiments involving bumble-bees, but the experiments were conducted according to ASAB/ABS guidelines.

## Decision letter and Author response

Decision letter https://doi.org/10.7554/eLife.31262.016
Author response https://doi.org/10.7554/eLife.31262.017

# Additional files

## Supplementary files

• Supplementary file 1: A summary of the temperature patterns observed on each of the 118 species thermographed, and the additional 18 cultivars and subspecies. Species are ordered taxonomically. The temperature at the hottest and coldest region of the flower and the difference in temperature between these points is also given. Plants derived from the same species where counted together for occurrence or average temperature difference calculations. Plants not used in the calculations are marked with a '*' next to their Δ temp. value.
DOI: https://doi.org/10.7554/eLife.31262.009

• Supplementary file 2: Hourly weather data is provided for each hour thermographs were collected. All weather data was obtained from the nearest Met Office weather station: for Bristol survey days, Filton weather station (51°31'15.6"N 2°34'33.6"W); for Botanic Garden of Wales survey days, Aberporth weather station (52°07'48.0"N 4°32'20.4"W).
DOI: https://doi.org/10.7554/eLife.31262.010

• Supplementary file 3: Daily weather data for days where sampling took place. All weather data was obtained from the nearest Met Office weather station: for Bristol survey days, Filton weather station; for Garden of Wales survey days Saron weather station (52°01'N 4°37'W) for daily temperature and rainfall data, Aberporth weather station for all other data.
DOI: https://doi.org/10.7554/eLife.31262.011

• Transparent reporting form
DOI: https://doi.org/10.7554/eLife.31262.012

## Major datasets

The following dataset was generated:

| Author(s) | Year | Dataset title | Dataset URL | Database, license, and accessibility information |
|---|---|---|---|---|
| Harrap MJM, Rands S A, Hempel de Ibarra N, Whitney H M | 2017 | Data from: The diversity of floral temperatue patterns, and their use by pollinators | https://doi.org/10.5061/dryad.qp244 | Available at Dryad Digital Repository under a CC0 Public Domain Dedication |

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
