## [Decision Letter]

Thank you for submitting your article "The diversity of floral temperature patterns, and their use by pollinators" for consideration by *eLife*. Your article has been reviewed by two peer reviewers, and the evaluation has been overseen by a Reviewing Editor and Ian Baldwin as the Senior Editor. The following individuals involved in review of your submission have agreed to reveal their identity: Danny Kessler (Reviewer #2); Klaus Lunau (Reviewer #3).

The reviewers have discussed the reviews with one another and the Reviewing Editor has drafted this decision to help you prepare a revised submission.

The two reviewers and the Reviewing Editor have read your article with much interest. The conclusion is that you have shown an interesting diversity in temperature patterns in flowers of many plant species. Regarding the diversity of plant species two important issues have been identified: 1- the species diversity seems lower than you suggest and 2- it is not clear whether these flowers represent bee-pollinated flowers.

Regarding the recorded temperature patterns there is agreement that the flowers represent different temperature patterns (even when a major part would be in difference between periphery and centre), but there are questions on how the bee behaviour was scored and statistically analysed. Based on this evaluation I wish to stimulate you to present thermal data on a sufficiently large proportion of the flowers that are known to be bee pollinated and provide effective response/revision to the questions on scoring and statistically analysing bee behaviour.

Based upon the revision the reviewers will make a decision on whether the revision has sufficiently improved the manuscript to warrant publication in *eLife*.

The detailed evaluations by the reviewers can be found below.

*Reviewer #2:*

Harrap et al. present compelling evidence that flowers of many plant species show temperature patterns. By thermographing flowers of 136 plant species or cultivars they show that flowers of most of the plants measured may gain within-flower temperature differences greater than 2°C if measured under sunlight conditions. Furthermore the authors are able to demonstrate that bumblebees are able to learn these kinds of floral temperature patterns if associated with a food source. The conducted bumblebee experiments are just as easy, as elegant, and show very convincing that bumblebees indeed could use temperature patterns within a flower as a floral cue to recognize and learn rewarding flowers. Especially the fact that different temperature patterns have been used simultaneously in one set of experiments is a very good and essential tool to convince the reader.

Although I like the overall presentation, the results, as well as the conclusions of the manuscript I have yet to criticize the presentation of the thermography data a little. The authors speak of flower species which is not wrong, but which leads automatically to the idea that 136 plant species have been measured. In fact, if I counted right, only 122 plant species have been observed, while in seven cases cultivars of the same species have been measured. It's very likely that these plant cultivars have equal properties, which is indeed reflected in Supplementary file 1. Most of these cultivars represent species that showed temperature patterns above 2°C difference, and thus influence the main result positively. Using several cultivars derived from one species e.g. Cistus or Gazania thus seems strange to me if each of these are counted as separate "flower species" and thus contributes to the overall result of 61%. On the other hand, as in the case of *Knautia macedonica*, one native variety differs highly from another, which is very interesting on its own. In short I would suggest keeping the table as it is, but adjust or add the statistics based on the plant species not the "flower species".

The manuscript also would benefit from some discussion on the ecological background. For example, are flowers of plants that grow in shady areas less likely to express these thermo-patterns, as the described floral pattern relies on direct sun light. Insects thus can use this cue only on sunny days in open spaces, which suggest that plant species growing on meadows or in more arid environments should have evolved much stronger probabilities to use sunlight in order to create thermal differences, than plants which grow hidden in the forest or live in "bad weather" environments. Another interesting question to add to the intro or discussion: Are all investigated "flower species" bee pollinated or can you see some trends if bee pollinated plants are more likely show the observed pattern?

None of these suggestions however weaken the main results, but are rather comments to improve the manuscript.

*Reviewer #3:*

The authors present two studies, one thermography study about temperature patterns in natural flowers and one behavioural study about the bumblebees' ability to discriminate between different temperature patterns of artificial flowers. Both studies are combined to a single take home message "Thus temperature patterns can be added to the growing number of floral cues that pollinators utilise to identify more rewarding flowers in their environment." In my opinion this far reaching conclusion is not justified by the data presented by the authors for the following reasons. The temperature patterns of the natural flowers analyzed were predominantly from species with radially symmetrical flowers which are not or only rarely visited and pollinated by bumblebees. The diversity of temperature patterns tested in the experiments with bumblebees did not mirror the diversity of temperature patterns found in natural flowers. The bumblebees sensed the temperature pattern via touch by walking around the artificial flowers, a corresponding behaviour in natural flowers has not yet been described to my knowledge.

The diversity of floral temperature patterns seems overestimated, since all flowers have a cooler center. The authors stated: "While the temperature patterns that pollinators may encounter can vary greatly between species, we must determine whether pollinators can use such differences in temperature patterns to inform foraging in order to show these differences are floral cues." The data show that the variation of the floral temperature patterns is mainly caused by the difference in temperature between the cool center and the warm periphery and to a far lesser extend by differences in the spatial patterns similar to those that have been used in the discrimination tests.

Moreover, the selected species do not represent a broad range of possible floral shapes, colours and phylogeny, but are mostly not bee-pollinated species with radially symmetrical flowers.

The scoring was done as follows: In the learning phase experiments feeding on a rewarding flower, or not feeding on a nonrewarding flower, was deemed a correct action. Doing otherwise was deemed incorrect. In the test phase all flowers were non-rewarding, therefore scoring flowers 'rewarding' and 'nonrewarding' was determined by the reward scheme in the preceding learning phase. I do not understand the scoring in the test phase: should feeding be replaced by probing?

Since the authors did not score revisits to the artificial flowers that have already been visited, the relation between rewarding and nonrewarding artificial flowers is changing. My question is whether and how this changed relation has been implemented in the statistical tests?

---

## [Author Response]

Reviewer #2:[…] Although I like the overall presentation, the results, as well as the conclusions of the manuscript I have yet to criticize the presentation of the thermography data a little. The authors speak of flower species which is not wrong, but which leads automatically to the idea that 136 plant species have been measured. In fact, if I counted right, only 122 plant species have been observed, while in seven cases cultivars of the same species have been measured. It's very likely that these plant cultivars have equal properties, which is indeed reflected in Supplementary file 1. Most of these cultivars represent species that showed temperature patterns above 2 °C difference, and thus influence the main result positively. Using several cultivars derived from one species e.g. Cistus or Gazania thus seems strange to me if each of these are counted as separate "flower species" and thus contributes to the overall result of 61%. On the other hand, as in the case of Knautia macedonica, one native variety differs highly from another, which is very interesting on its own. In short I would suggest keeping the table as it is, but adjust or add the statistics based on the plant species not the "flower species".

This is a very valid observation, and we thank the reviewer for pointing it out! The manuscript has been amended as the reviewer suggests. Cultivars and subspecies derived from the same plants are now counted together for calculation of the aforementioned statistics, with the lowest Δtemp (i.e. the most conservative) value used to calculate average temperature difference. Accordingly we refer to 118 flowering species now (Abstract, main text and Results). In addition, the following changes add clarification:

·

Subsection “Sampling of floral temperature patterns”: we have inserted clarification of how cultivars and subspecies were used in calculation of statistics;

The Table in Supplementary file 1 now indicates which floral thermographs were excluded from calculations of temperature pattern statistics – see the legend for this file;

Additionally, a comment drawing attention to the variation between temperature patterns of subspecies and cultivars derived from the same species has been added to the second paragraph of the Discussion.

The manuscript also would benefit from some discussion on the ecological background. For example, are flowers of plants that grow in shady areas less likely to express these thermo-patterns, as the described floral pattern relies on direct sun light. Insects thus can use this cue only on sunny days in open spaces, which suggest that plant species growing on meadows or in more arid environments should have evolved much stronger probabilities to use sunlight in order to create thermal differences, than plants which grow hidden in the forest or live in "bad weather" environments.

A paragraph has been added (Discussion, third paragraph), commenting on how such ecological aspects might be expected to affect the capacity of plants to produce temperature patterns and how that may relate to their evolution.

Another interesting question to add to the intro or discussion: Are all investigated "flower species" bee pollinated or can you see some trends if bee pollinated plants are more likely show the observed pattern?

Edits have been made to the manuscript to clarify that sampling of floral thermal patterns was not limited to just ‘bee-flowers’, and clarifying why bumblebees were appropriate for conditioning experiments. These include:

The last paragraph of the main text has been restructured to include some clarity on how the survey relates to bee experiments;

Subsection “Diversity of floral temperature patterns”, last paragraph, changed ‘that bees’ to ‘at least bees’;

Subsection “Bumblebees discriminate between flowers with different temperature patterns”, first paragraph. A brief discussion of why bumblebees were a good choice for conditioning experiments with temperature patterns and how their visitation habits relate to the diversity of flowers sampled;

Starting ‘It did not appear…’: A comment is added to the first paragraph of the Discussion noting that there did not appear to be a clear trend within the sampled temperature patterns were associated with a particular pollinator group. We have also related this to the possibility temperature pattern detection is not limited to bees;

Subsection “Sampling of floral temperature patterns”, comments added to the criteria of selection to demonstrate we were aiming to survey flowers visited by many pollinators.

None of these suggestions however weaken the main results, but are rather comments to improve the manuscript.

We very much agree the manuscript is now clearer and strongly benefits from these edits – many thanks!

Reviewer #3:The authors present two studies, one thermography study about temperature patterns in natural flowers and one behavioural study about the bumblebees' ability to discriminate between different temperature patterns of artificial flowers. Both studies are combined to a single take home message "Thus temperature patterns can be added to the growing number of floral cues that pollinators utilise to identify more rewarding flowers in their environment." In my opinion this far reaching conclusion is not justified by the data presented by the authors for the following reasons.

Changes to our concluding statements have been made and hopefully this reads better now (see Discussion, last paragraph).

The temperature patterns of the natural flowers analyzed were predominantly from species with radially symmetrical flowers which are not or only rarely visited and pollinated by bumblebees. The diversity of temperature patterns tested in the experiments with bumblebees did not mirror the diversity of temperature patterns found in natural flowers.

The reviewer’s comments suggest that we have not been clear enough. The temperature pattern survey aimed to demonstrate more broadly the fact that a diversity of temperature patterns across flowers can be found and that these are flowers not only differed in colour, shape and size but also are visited by a range of pollinators. It could well be that bumblebees are not seen on some of the flower species frequently, but they often forage on radial-symmetric flowers, are generalist foragers visiting and adjusting to a wide range of flower available, and are good learners. Thus, they are a good choice to conduct the relevant behavioural experiments to demonstrate that pollinators have the capacity to utilise these patterns as a cue. Furthermore, we have seen in the gardens that bumblebees visited many of the flowers that were surveyed which shows that they are not deterred from foraging on flowers with such temperature patterns. Clarifying edits include:

The last paragraph of the main text has been restructured to make it clearer how the survey relates to bee experiments;

Subsection “Diversity of floral temperature patterns”, last paragraph, added ‘at least’. To emphasise that not only bee flowers were sampled;

Starting ‘It did not appear[…]’: A comment is added to the first paragraph of the Discussion noting that there did not appear to be a clear trend within the sampled temperature patterns were associated with a particular pollinator group. We have also related this to the possibility that temperature pattern detection is not limited to bees;

Subsection “Sampling of floral temperature patterns”, comments added to the criteria of selection to demonstrate we were aiming to survey flowers visited by many pollinators.

We have additionally added comments throughout the manuscript clarifying how bumblebee visitation habits relate to the diversity of flowers sampled and why bumblebees, particularly *B. terrestris*, was an appropriate choice of pollinator to investigate whether any pollinator can respond to the observed diversity of temperature patterns. Edits made are:

Main text, last paragraph, some context added for why bumblebees were tested added to summary;

Subsection “Bumblebees discriminate between flowers with different temperature patterns”, first paragraph: A section is added explaining how bees fit within the diversity of flower species sampled and their appropriateness for the pollinator to begin tests of the diversity of patterns observed.

Discussion, first paragraph: Comments on how several of the flower species with the most contrasting temperature patterns might relate to bumblebees.

Alteration and clarification in reference to how the temperature patterns from conditioning experiments relate to the natural diversity of temperature patterns is discussed below.

The bumblebees sensed the temperature pattern via touch by walking around the artificial flowers, a corresponding behaviour in natural flowers has not yet been described to my knowledge.

The reviewer is correct in that, existing evidence clearly suggests that bees need to make contact with objects to sense its temperature, similarly to humans. However, the bees do not walk around on the artificial flowers. To better explain how the bee typically interacted with the flowers, we have added further details about it in the first paragraph of the subsection “Learning experiments”. The landings we observed were not different from those seen on real flowers.

The diversity of floral temperature patterns seems overestimated, since all flowers have a cooler center. The authors stated: "While the temperature patterns that pollinators may encounter can vary greatly between species, we must determine whether pollinators can use such differences in temperature patterns to inform foraging in order to show these differences are floral cues." The data show that the variation of the floral temperature patterns is mainly caused by the difference in temperature between the cool center and the warm periphery and to a far lesser extend by differences in the spatial patterns similar to those that have been used in the discrimination tests.

The reviewer is correct that differences between the centre and periphery make up a greater part of the diversity of temperature patterns observed, as stated in the first paragraph of the subsection “Diversity of floral temperature patterns”. However, it is not the case that most flowers have colder centres as the review states. In fact most flowers show a predominately hot centre, reflected in Figure 1. Consequently, we have added a comment clarifying that the differences of hot and cold centres made up a large part of the temperature pattern diversity, with examples from Figure 1.

We realise that describing the artificial temperature patterns by the arrangement heat sinks (‘cross’, ‘bar’ and ‘circle’), while easier to follow, makes it unclear how they relate to natural temperature patterns. We thus relate more clearly how the artificial flowers relate to the natural diversity of temperature patterns more directly when describing them for the first time. Edits include:

Subsection “Bumblebees discriminate between flowers with different temperature patterns”, second paragraph is expanded to include a description of temperature patterns and how they relate to natural patterns;

Discussion, first paragraph, we have added comments contextualising the results and how artificial flowers related to natural flowers and what this tells us about temperature patterns detection.

Moreover, the selected species do not represent a broad range of possible floral shapes, colours and phylogeny, but are mostly not bee-pollinated species with radially symmetrical flowers.

The thermographic survey we present presents a range of differing patterns across a wide range of flowering plants, mostly constrained by the range of blooming species that were readily available – we demonstrate that there is diversity available to pollinators that may be visiting these flowers. We then focus on bumblebees as a model species for exploring whether a known pollinator is able to differentiate heat patterns in a feasible form of flower. By demonstrating that this is possible, what we present is a demonstration that heat patterns may be important for pollinators (of which *B. terrestris* is a single example), rather than argue that all the patterns surveyed are important specifically for bumblebees. We hope that the editing of language throughout the manuscript has made our intention here more clear:

Subsection “Sampling of floral temperature patterns”, has been edited;

Subsection “Sampling of floral temperature patterns”, a comment has been added explaining why, due to some limitations of IRT cameras some very small floral blooms were not sampled.

The scoring was done as follows: In the learning phase experiments feeding on a rewarding flower, or not feeding on a nonrewarding flower, was deemed a correct action. Doing otherwise was deemed incorrect. In the test phase all flowers were non-rewarding, therefore scoring flowers 'rewarding' and 'nonrewarding' was determined by the reward scheme in the preceding learning phase. I do not understand the scoring in the test phase: should feeding be replaced by probing?

The reviewer has picked up a valid point here. ‘Probing’ might be a better term to describe some of the bees’ behaviour in the tests, but also during the training when mistakenly extending the proboscis towards the water solution in non-rewarding flowers. Bees only really fed on rewarding flowers. As advised we have altered the language used in the manuscript (now making use of term ‘probing’) to reduce confusion. In the third paragraph of the subsection “Bumblebees discriminate between flowers with different temperature patterns”, we have also altered the phrasing of our criteria of foraging success.

Since the authors did not score revisits to the artificial flowers that have already been visited, the relation between rewarding and nonrewarding artificial flowers is changing. My question is whether and how this changed relation has been implemented in the statistical tests?

This is correct, the balance changes as the flowers get emptied by a bee during a foraging bout. However, the changes are small. For instance, in the large flower experiments flowers were refilled carefully during a bout. In the small flower experiments the bee could encounter more empty flowers because they are not refilled. However, in the small-flower experiments bees can fill up without having to visit all flowers and return to the nest. After that flowers are refilled to prepare the next foraging bout. Bees are resilient to encountering an empty flower that they have experienced to be rewarding, as known from numerous foraging and learning studies, but also as demonstrated here by the fact that bees increased their correct choices over time. We did not include the responses of bees when they revisited a flower for two reasons. First, if the flower was of the rewarding type it would be empty. The empty well in itself might preclude a response, and this would be indistinguishable from an error of not responding to the rewarding flower type. Second, if the repeatedly visited flower was of the non-rewarding type, it would not be empty and thus simply different in its presentation from the repeatedly visited rewarding flower. We have added clarification to the text:

Subsection “Learning experiments, second paragraph, clarification of the reasoning for avoiding interference with the flowers while the bee was foraging where possible.

Subsection “Statistical analysis”, the third paragraph added to explain this potential influence and our reasoning for exclusion from the analysis.